# Zika virus modulates human fibroblasts to enhance transmission success in a controlled lab-setting
Raimondas Mozūraitis [1,2,3,13], Karsten Cirksena [4,13], Mohammad Raftari[1,5], Melika Hajkazemian[1], Musa Mustapha Abiodun[1], Juela Brahimi[1,6], Sandra Radžiutė[2], Violeta Apšegaitė[2], Rasa Bernotienė[7], Lech Ignatowicz[6], Tessy Hick[8], Andreas Kirschning[9], Annasara Lenman[8], Gisa Gerold [4,8,10,11,13] ✉ & S. Noushin Emami[1,5,6,12,13] ✉

Transmission of Zika virus (ZIKV) has been reported in 92 countries and the geographical spread of invasive virus-borne vectors has increased in recent years. Arboviruses naturally survive between vertebrate hosts and arthropod vectors. Transmission success requires the mosquito to feed on viraemic hosts. There is little specific understanding of factors that may promote ZIKV transmission-success. Here we show that mosquito host-seeking behaviour is impacted by viral infection of the vertebrae host and may be essential for the effective transmission of arboviruses like ZIKV. Human skin fibroblasts produce a variety of metabolites, and we show that ZIKV immediately alters gene/protein expression patterns in infected-dermal fibroblasts, altering their metabolism to increase the release of mosquito-attractive volatile organic compounds (VOCs), which improves its transmission success. We demonstrate that at the invasion stage, ZIKV differentially altered the emission of VOCs by significantly increasing or decreasing their amounts, while at the transmission stage of the virus, all VOCs are significantly increased. The findings are complemented by an extensive meta-proteome analysis. Overall, we demonstrate a multifaceted role of virus-host interaction and shed light on how arboviruses may influence the behaviour of their vectors as an evolved means of improving transmission-success.

The human body comprises around 40 trillion ($3.7 \times 10^{13}$) cells[1], which produce an enormous variety of metabolites including volatile organic compounds (VOCs). The qualitative and quantitative changes in VOC profiles reflect the optimum status of healthy cells. The skin is the body's largest organ, serving multiple essential functions, including acting as a physical barrier for protection, a site for sensory perception, and a centre for vitamin synthesis. The release of VOCs through the skin, which contributes to the distinct odours of the human body, is a familiar part of our daily experience[2]. These VOCs include a large number of volatiles that can be listed as carboxylic acids, aldehydes, alcohols or ketones[3]. Dermal fibroblasts are the main cell type present in skin connective tissue (dermis)[4]. These mesenchymal/stromal cells derived from the embryonic mesoderm, reside in the dermal layer of skin. They produce extracellular matrix proteins to strengthen the dermal compartment and interact with epidermal cells[5].

Zika virus (ZIKV) had not been widely known until a series of outbreaks occurred with severe clinical complications that made it a matter of global public health concern. The emergence of ZIKV followed a typical pattern of a vector-borne disease being introduced into a new ecosystem and

[1]Department of Molecular Biosciences, Wenner-Gren Institute, Stockholm University, Stockholm, Sweden. [2]Laboratory of Chemical and Behavioral Ecology, Institute of Ecology, Nature Research Centre, Vilnius, Lithuania. [3]Department of Zoology, Stockholm University, Stockholm, Sweden. [4]Institute for Biochemistry & Research Center for Emerging Infections and Zoonoses (RIZ), University of Veterinary Medicine Hanover, Hanover, Germany. [5]Vector Biology Department (VBD), Liverpool School of Tropical Medicine (LSTM), Liverpool, UK. [6]Molecular Attraction AB, Anderstorpsvägen 16, Solna, Stockholm, Sweden. [7]Laboratory of Entomology, Institute of Ecology, Nature Research Centre, Vilnius, Lithuania. [8]Department of Clinical Microbiology, Umeå University, Umeå, Sweden. [9]Institute of Organic Chemistry, Leibniz University Hannover, Schneiderberg 1B, Hannover, Germany. [10]Wallenberg Centre for Molecular Medicine (WCMM) & Department of Clinical Microbiology, Virology, Umeå University, Umeå, Sweden. [11]Institute of Virology, Medical University of Innsbruck, Innsbruck, Austria. [12]Natural Resources Institute, FES, University of Greenwich, London, UK. [13]These authors jointly supervised this work: Raimondas Mozūraitis, Karsten Cirksena, Gisa Gerold, S. Noushin Emami. ✉e-mail: gisa.gerold@i-med.ac.at; noushin.emami@lstmed.ac.uk

host population and spreading rapidly with severe implications for human health. ZIKV is a mosquito-borne virus belonging to the genus *Flavivirus* and is transmitted to humans by mosquitoes of the genus *Aedes*. Both *Aedes aegypti* and *Aedes albopictus* are the primary vectors for ZIKV transmission in nature[6]. Vector-mediated transmission of ZIKV is initiated when a blood-feeding female *Aedes* mosquito injects the virus into the skin of its mammalian host, followed by infection of a wide range of permissive cells. Indeed, skin cells, including dermal fibroblasts were found to be permissive to ZIKV infection. Infection of skin fibroblasts rapidly resulted in the presence of high RNA copy numbers and a gradual increase in the production of ZIKV particles over time, indicating an active viral replication stage. In humans, the incubation period from mosquito bite to symptom onset is ≈3–12 days[7]. Accumulating data indicate that ZIKV alters the biochemical processes of the infected cells by modifying glucose[8,9] and fatty acid[10] metabolism. However, no published data has yet revealed any changes in the composition of infected cell volatome. While Zhang and colleagues[11] demonstrated that flaviviruses can manipulate host skin microbiota to produce an odour that attracts mosquitoes, the specific role of infected human skin cells in manipulating mosquitoes' behaviour remains unclear.

Mosquitoes have made themselves at home in new geographical regions throughout recent years, bringing with them some historically exotic diseases. Epidemiologic and laboratory studies have implicated various *Aedes* spp. mosquitoes as ZIKV vectors[11]. In mosquitoes, it appears that the viral load is initially high on the day of feeding, but then decreases to undetectable levels for about ten days. After this incubation period, the viral content increases again by day 15 and remains high from days 20 to 60. This is important because it suggests that it takes about 10 days for the virus to reach the salivary glands of the mosquito where it can potentially be transmitted to humans[7]. There is currently no specific treatment or vaccine available to mitigate or prevent ZIKV infection. Prevention measures, particularly conventional vector control, are currently the priority while we await these and other advances in control of the substantial harm caused by this virus. The World Health Organization has issued recommendations on this matter[12].

To further understand this complex issue, this study aims to demonstrate how ZIKV manipulate vector behaviour by altering gene/protein expression in human dermal fibroblasts at different stages of infection. These changes lead to an increased release of mosquito-attractive VOCs, ultimately enhancing transmission success. Here, we describe how ZIKV infection of human host cells provoked the modified feeding behaviour of its tiger mosquito vector in a manner that plausibly results in enhanced transmission success.

## Results

### ZIKV alters the volatome of human dermal fibroblasts

Given that human dermal fibroblasts are among the first cells to be infected by ZIKV, it is crucial to understand how the virus alters the production of VOCs. Since VOCs can influence mosquito behaviour, changes in the VOC profile could play a significant role in facilitating the spread of the virus. We found that ZIKV infection altered the volatome of human dermal fibroblasts (Supplementary Fig. 1a, c, Supplementary Table 1, and Supplementary Data file 1). The comparison of VOC profiles sampled from cell cultures infected with ZIKV at the invasion stage versus uninfected cells revealed that the release of fourteen VOCs was significantly increased. The largest increase was recorded for 2,4-dimethyl-1-heptane followed by branched hydrocarbons (Supplementary Data file 1:58, 60, 64, 65), 4-methyl-heptane, dodecanal, and sulcatone. The emission of five VOCs significantly decreased with the largest change registered for branched hydrocarbon (Supplementary Data file 1: 44, Supplementary Table 1 and Supplementary Fig. 1a, b). Interestingly, at the transmission stage of the virus, amounts of all eight significantly altered VOCs were increased compared to emissions from healthy cells. The largest changes were observed for sulcatone, decanal, dodecanal, and 2-methyl-1-pentene (Fig. 1a–e, Supplementary Table 1b, and Supplementary Fig. 1c, d).

### ZIKV odour enhances host-seeking and feeding behaviour of its vector

The transmission success of ZIKV is closely linked to the behaviour of its mosquito vectors, particularly their ability to locate and feed on hosts. Understanding how ZIKV-induced changes in odour influence mosquito host-seeking and feeding behaviour is crucial for identifying the mechanisms by which the virus enhances its transmission potential. To study the ZIKV-related odour effects on mosquito attraction, we tested six commercially available synthetic VOCs analogous to the compounds emitted increasingly from virus infected cells during the two different stages of the virus replication cycle (invasion and transmission). The synthetic VOCs elicited electroantennographic responses in mated five to seven days old *Ae. aegypti* females (Supplementary Fig. 2). Using a Y-tube olfactometer, we demonstrated that olfactory active volatiles tested in the blends representing emissions from fibroblasts infected with ZIKV at invasive (invasion blend) and transmission (transmission blend) stages significantly attracted *Ae. aegypti* females. In contrast, *Anopheles gambiae* sensu lato (s.l.) mosquitoes, which do not support ZIKV transmission naturally and are highly efficient at transmitting the *Plasmodium* parasites that cause malaria, did not show a preference for the blends versus the control (Fig. 1f–h). We conclude that ZIKV modulatory effects within the infected human host skin cells, affect the blood-seeking of its vector, *Ae. aegypti* mosquito.

We observed a, striking propensity of female *Ae. aegypti* mosquitoes to be attracted and land on membrane feeders nearby paraffin oil containing ZIKV-related synthetic VOC blends (Fig. 2a–i). 3D videography assays of mosquitoes in wind tunnel experiments revealed that all six VOCs released from paraffin oil as single compounds attracted significantly more females compared to control bearing paraffin oil only. Dodecanal, 2-methyl-1-pentene, and 4-methylheptane showed the highest efficiency. We also showed that the transmission blend comprised of four synthetic VOCs, which are released at a ratio similar to that determined from the cell culture bearing ZIKV at the transmission stage, was significantly more attractive compared to five components invasion blend (Fig. 2b–i and Supplementary Table 2a).

The invasion and transmission blends modified the blood-feeding behaviour of mated five to seven days old *Ae. aegypti* females, as the amount of blood meal obtained by females were more than doubled and tripled when fed on blood with nearby located invasion and transmission blends, respectively, compared to control blood in the absence of ZIKV related-odour (Fig. 3a, b and Supplementary Table 2b). The mean amount of hematin excreted by females three days post-feeding was around 6 μg/ml, and 12 μg/ml for the group of females exposed to invasion and transmission blends, respectively compared to the control with no VOCs released (~ 2 μg/ml). The amount of hematin excreted by mosquitoes directly correlates to the amount of blood consumed. Taken together, these findings suggest that the blood with the ZIKV-related odour provokes feeding behaviour associated with the taking of substantially larger blood meal volumes by *Aedes* vectors.

To further decipher the feeding behaviour of invasion and transmission ZIKV-related blends, we provided healthy human blood meals with and without ZIKV related-odour to mosquitoes and examined the percentage of females that landed and initiated probing and feeding (referred to as accumulated blood feeding proportion % within 800 s) (Fig. 3c–e). Approximately 100% of the mosquitoes displayed behavioural persistence (an increased proportion of female mosquitoes landed and fed on membrane feeders) when provided with ZIKV-related odour nearby blood meal (invasion and transmission blends) compared to 40% of mosquitoes provided with blood alone (with nearby placed paraffin oil without synthetic VOCs) during 800 s (Fig. 3d, e). ZIKV odour could potentially benefit the virus transmission rate to the mosquito due to easier detection of the blood source location in the short time of feeding. Ten seconds after the start of the feeding experiment, we observed that 10% and 27% of mosquitoes were registered on the membrane feeders with nearby placed invasion and transmission blends, respectively, compared to 3% of females feeding on the control feeder (without nearby

 

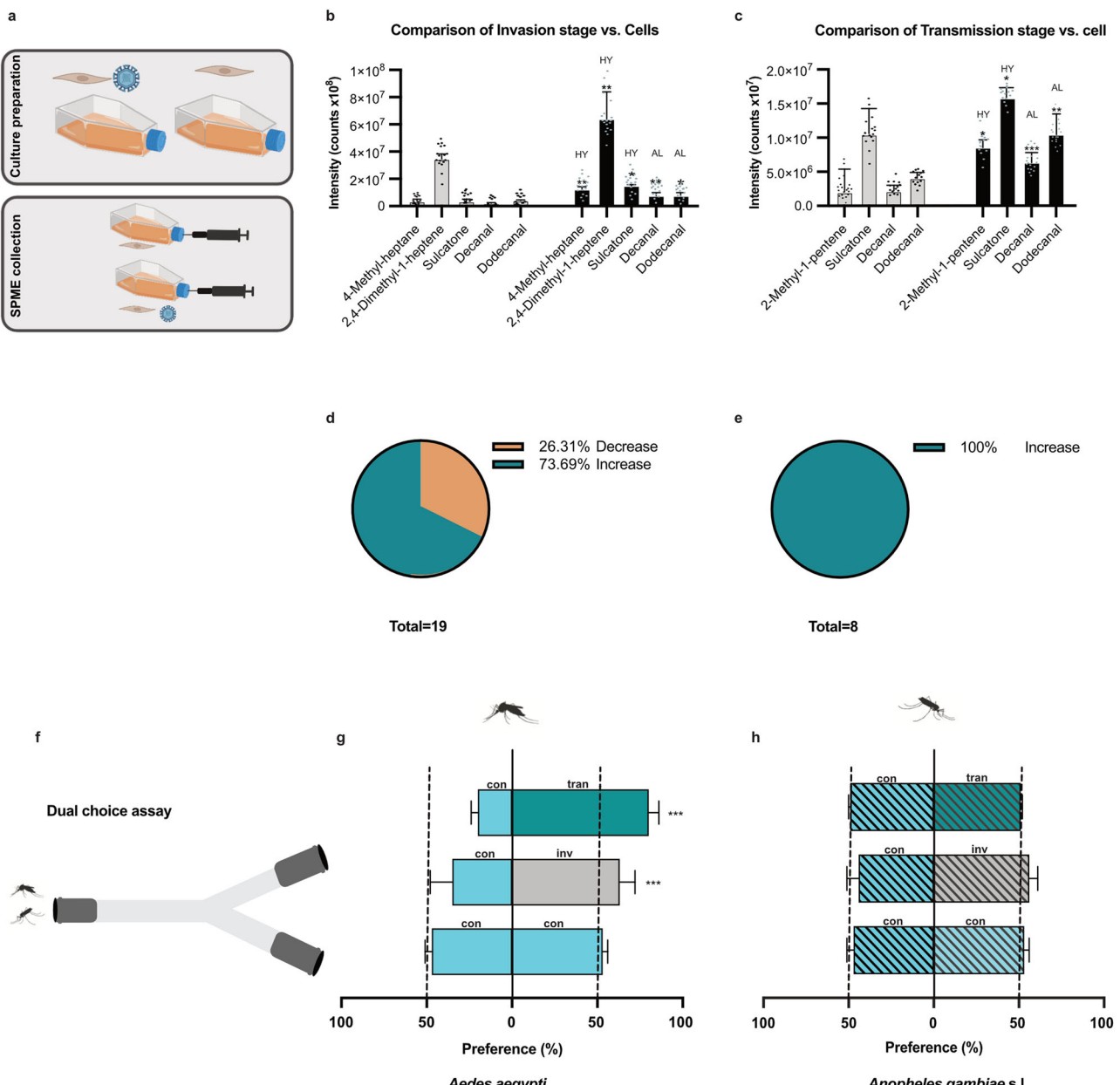

**Fig. 1 | Volatile organic compounds (VOCs) sampled in vitro from fibroblast tissue culture infected with the ZIKV at invasion and transmission stages and their effect on mosquito attraction. a** Schematic model of study for virus culturing and solid phase microextraction (SPME) collection. **b** Comparison of fibroblasts' VOCs carrying the invasion stage of the ZIKV with uninfected fibroblasts' VOCs using GC-MS [enhanced VOCs: 4-Methyl heptane: t = 1.14, $p = 0.002$; 2,4-Dimethyl-1-heptene: t = 5.43, $p = 0.003$; Sulcatone: t = 3.65, $p = 0.04$; Decanal: t = 2.80, $p = 0.006$; Dodecanal: t = 2·49, $p = 0·01$; grey bars represented cells without virus infection and black bars showed cells infected with virus in invasion stage]. **c** Comparison of fibroblasts' VOCs carrying the transmission stage of the ZIKV with uninfected fibroblast cells' VOCs using GC-MS [enhanced VOCs: 2- Methyl-1-pentene: t = 1.71, $p = 0.046$; Sulcatone: t = 2.17, $p = 0.03$; Decanal: t = 3.63, $p < 0.001$; Dodecanal: t = 3.10, $p = 0.002$; grey bars represented cells without virus infection and black bars showed cells infected with virus in transmission stage]. **d, e** The total percentage of VOCs that significantly increase or decrease as a result of the proliferation of the virus in cells at the invasion or transmission stage in comparison with uninfected cells (%). **f** Schematic model of study for effects of ZIKV related-blends (invasion and transmission) on mosquito attraction in a dual choice attraction assay. **g** *Ae. aegypti* preference for blend solvent (Control), ZIKV related-invasion blend (Invasion) or ZIKV related-transmission blend (Transmission) in a dual choice attraction assay [Control: $\chi^2_1 = 0.26$, $p = 0.60$; Invasion: $\chi^2_1 = 4.38$, $p < 0.001$; Transmission: $\chi^2_1 = 14.09$, $p < 0.001$]. **h** *An. gambiae* s.l. preference for solvent (Control = paraffin oil), ZIKV invasion blend (Invasion) or ZIKV related-transmission blend (Transmission) in a dual choice attraction assay [Control: $\chi^2_1 = 2.43$, $p = 0·11$; Invasion: $\chi^2_1 = 3.22$, $p = 0.07$; Transmission: $\chi^2_1 = 2.44$, $p = 0.12$]. Significant effects were determined using GLMM models (lmer) [in all experiments: $n = 30$/experimental replication]; Bars represented by $\beta$-estimation generated by the mixed model ± SE ($\beta$-lmer ± SE); asterisks denote significant differences (*$p < 0.05$; **$p < 0.01$; ***$p < 0.001$; ns, non-significant).

located synthetic VOCs). These findings all point to virus-provoked odour modulatory effects within the ZIKV infected human cells, affecting the blood-seeking and feeding behaviour of its vector, *Ae. aegypti* mosquito. We also showed that the presence of ZIKV-related odour at invasion or transmission stages significantly increased the mosquito blood meal size (Fig. 3b), which was independent of mosquito body size. Interestingly, both mosquito fecundity and survival were also enhanced by the ZIKV induced odour (Fig. 3f, g).

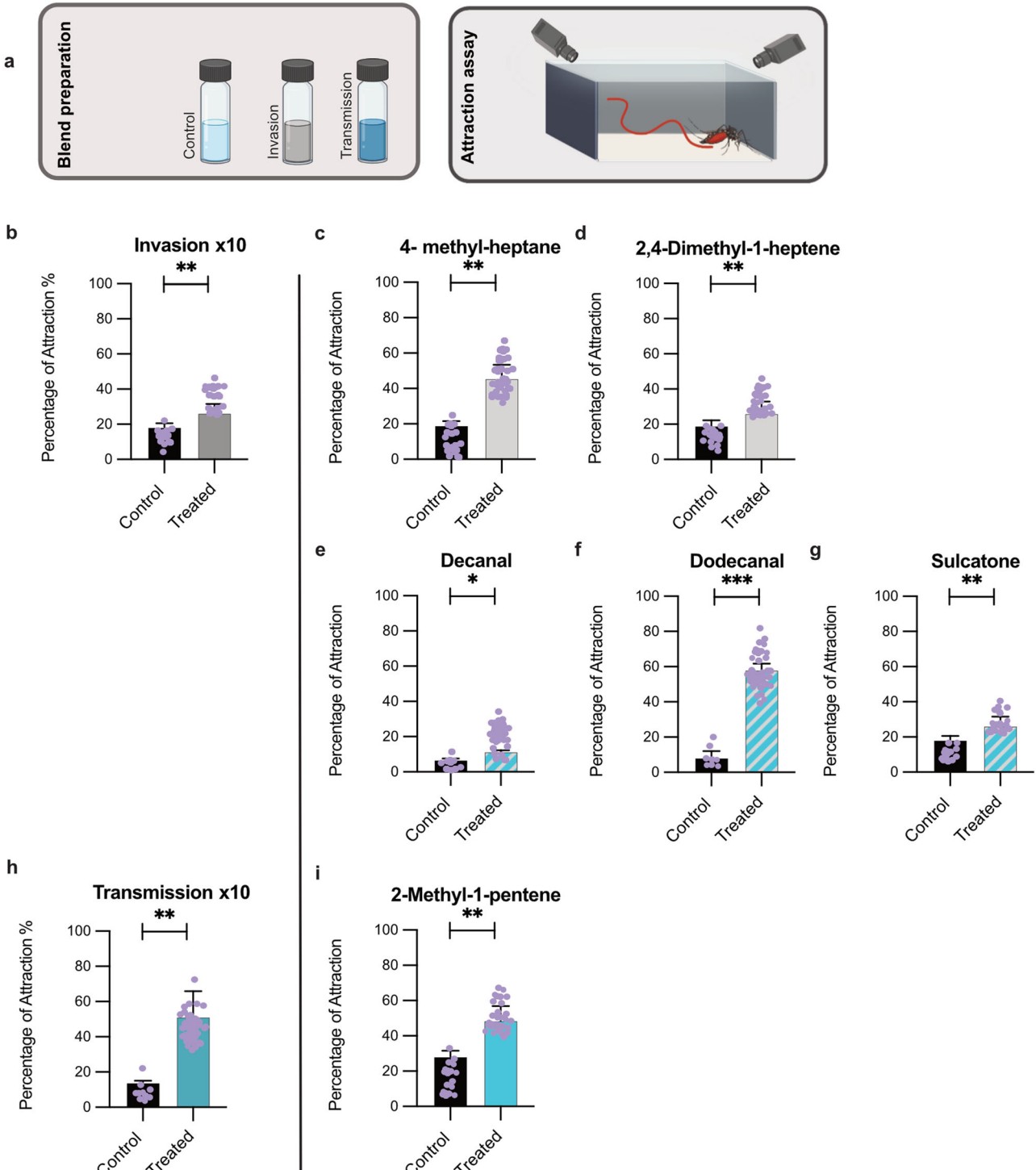

**Fig. 2 | Behavioural responses of *Ae. aegypti* females to synthetic odours singularly and as blends corresponding to those released from ZIKV infected fibroblast tissue culture at two different stages, invasion and transmission.**
**a** Schematic model of invasion and transmission blends' effects on mosquito host-seeking behaviour using a free-flight arena and 3D-tracking system. **b** Attraction of *Ae. aegypti* to ZIKV related-VOCs of invasion blend [×10 concentration: $\chi^2 = 7.21$, $p = 0.007$]. **c, d** Attraction of *Ae. aegypti* to single compounds, 4-Methylheptane [$\chi^2 = 9.72$, $p = 0.002$] and 2,4-Dimethyl-1-heptene [$\chi^2 = 7.20$, $P = 0.007$] the amounts of these two compounds increased significantly in the invasion blend of the ZIKV compared to those in uninfected cells. **e–g** Attraction of females to single compounds, which amounts increased significantly in both invasion and transmission blends (ZIKV VOCs at invasion and transmission stages) compared to those in uninfected cells, Decanal [$\chi^2 = 6.02$, $p = 0.01$], Dodecanal [$\chi^2 = 15.45$, $p < 0.001$], and Sulcatone [$\chi^2 = 7.20$, $p = 0.007$]. **h** Attraction of *Ae. aegypti* to ZIKV related-VOCs of transmission blend [×10 concentration: $\chi^2 = 0.14$, $p = 0.001$]. **i** Attraction of *Ae. aegypti* to a single compound, 2- Methyl-1-pentene [$\chi^2 = 8.9$, $p = 0.003$], which amounts increased significantly only in the transmission blend compared to those in uninfected cells. These experiments were done in triplicate (random effect). Significant effects were determined using GLMM models (lmer) [in each experimental replicate: $n = 30$: In total $n = 90$]; Bars represented by $\beta$-estimation generated by the mixed model ± SE ($\beta$-lmer ± SE); asterisks denote significant differences (*$p < 0.05$; **$p < 0.01$; ***$p < 0.001$; ns, non-significant).

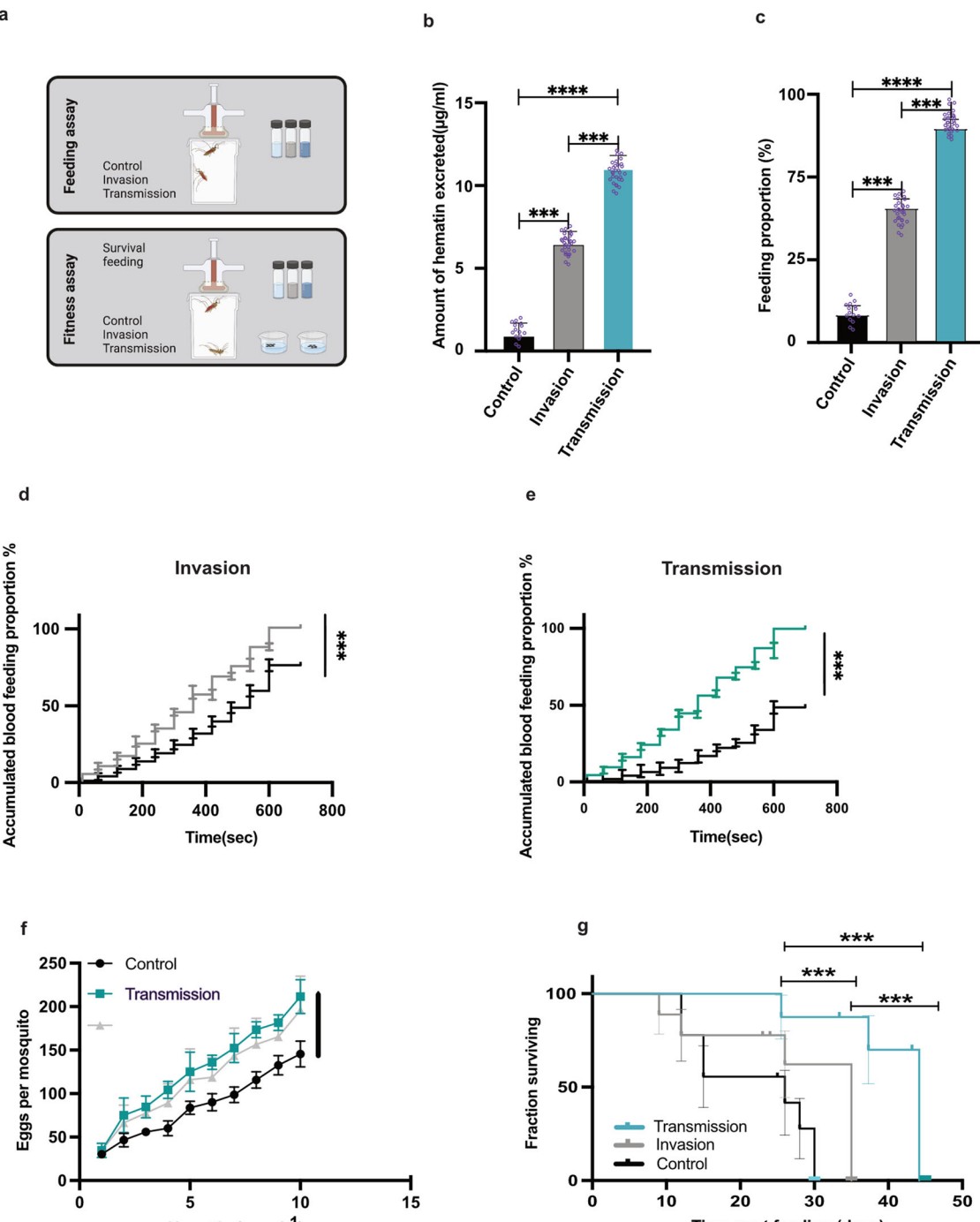

**Fig. 3 | Effects of bloodmeal with and without nearby ZIKV related-odour (invasion or transmission blends) on mosquito feeding behaviour and fitness.** **a–g** Females were fed on bloodmeal by membrane feeder without (control) or with nearby placed ZIKV related-odour, invasion blend or transmission blend. **a** Schematic model of experiments. **b** Average meal size was determined by the amount of hematin excreted and normalized individually to wing length [$\chi^2_1 = 50.60$, $p < 0.001$; Control vs Invasion $z = 3.15$, df = 1, $p < 0.001$; Invasion vs Transmission $z = 4.62$, df = 1, $p < 0.001$; Control vs Transmission $z = 7.01$, df = 1, $p < 0.001$]. **c** Blood feeding proportion (%) after exposure to feeders impregnated with the ZIKV related-odours at invasion [Control vs Invasion $z = 4.25$, df = 1, $p < 0.001$] and transmission [Invasion vs Transmission $z = 4.59$, df = 1, $p < 0.001$; Control vs Transmission $z = 6.59$, df = 1, $p < 0.001$] stages in comparison to control [$\chi^2_1 = 106.62$, $p < 0.001$]. **d** Accumulated blood feeding proportion (%) of mosquitoes was monitored per 10 seconds (Sec) at the first minute of the experiment and then continued per minute on feeders with/without nearby ZIKV related-odour at the invasion stage (invasion blend) until 800 Sec, [Cox hazard proportional model,

$\chi^2_1 = 217.50$, $p < 0.0001$]. **e** Accumulated blood feeding proportion (%) was monitored per 10 seconds (Sec) at the first minute of the experiment and then continued per minute on feeders with/without nearby ZIKV related-odour at transmission stage until 800 Sec, [Cox hazard proportional model, $\chi^2_1 = 22.68$, $p < 0.0001$]. **f** The relationship between fecundity (number of eggs per mosquito) and blood meal size (hematin) [treatment*hematin: $\chi^2_2 = 6.27$, $p < 0.001$; Invasion vs Transmission: $\chi^2_1 = 0.19$, $p = 0.66$; Control vs Transmission/Invasion: $\chi^2_1 = 6.91$, $p = 0.009$/ $\chi^2_1 = 7.83$, $p = 0.005$] of individual mosquitoes. **g** Survival of mosquitoes was monitored daily post feeding until natural death, [Cox hazard proportional model, $\chi^2_1 = 100.71$, $p < 0.001$; Invasion vs Transmission: $\chi^2_1 = 12.19$, $p < 0.001$; Control vs Transmission: $\chi^2_1 = 114.22$, $p < 0.001$; Control vs Invasion: $\chi^2_1 = 15.09$, $p < 0.001$]. These experiments were done in triplicate (random effect). Significant effects were determined using GLMM models (lmer) or Cox hazard proportional model [in all experiments: $n = 30$/experimental replication]. Bars represented by $\beta$-estimation generated by the mixed model ± SE ($\beta$-lmer ± SE); asterisks denote significant differences (*$p < 0.05$; **$p < 0.01$; ***$p < 0.001$).

## ZIKV manipulates host cell's transcriptome & proteome to facilitate its transmission

Understanding how ZIKV manipulates host cellular networks is essential for comprehending the virus's transmission dynamics. This knowledge could reveal innovative targets for intervention, potentially leading to the development of more effective strategies for preventing and controlling ZIKV spread by disrupting its ability to alter host cell functions and enhance its transmission. To gain an overall view of the initial effects of ZIKV on human dermal fibroblast cell transcription, Illumina sequencing was performed on whole cell RNA extracts from ZIKV infected human fibroblasts during invasion stage (10 h post-infection (hpi)) and transmission stage (24 hpi). RNAseq analysis revealed a distinct clustering of replicates of both, different conditions, and individual time points (Supplementary Fig. 3a and Supplementary Data file 2: RNAseq analysis). Of 11,535 transcripts quantified across all conditions, the expression of 401 transcripts were significantly affected (fold change ≥2, $q \leq 0.05$) by infection or the course of time with the largest fold changes observed 24 hpi with ZIKV (Fig. 4a). Cluster analysis of these transcripts revealed one cluster of transcripts affected by infection with ZIKV and two clusters of transcripts affected by the course of time independent from ZIKV infection (Fig. 4b and Supplementary Fig. 3b). Enrichment analysis among the 18 and 109 transcripts affected by ZIKV 10 hpi and 24 hpi, respectively (Supplementary Fig. 3c, d), mainly pointed to antiviral defence mechanisms (Fig. 4c). In addition, enrichment of transcripts involved in lipoprotein metabolic processes was observed with strong interconnection to processes like cholesterol and steroid metabolism (Fig. 4d). Among these transcripts we identified the apolipoprotein L gene family including, APOL1, APOL2 and APOL6 with up to a fivefold increase of expression (q value ≤ 0.05) after infection with ZIKV. However, significantly altered expression of these transcripts was only observed at 24 hpi (Fig. 4e–g).

Since protein expression during RNA virus infection is not only regulated at the level of transcription, we set out to identify changes in metabolic enzymes on the protein level that could lead to VOC production upon ZIKV infection. We compared the proteome of ZIKV- and Mock-infected immortalised human dermal fibroblasts applying liquid chromatography–mass spectrometry (LC-MS) analyses (Supplementary Data file 3I–VIII). However, observed changes in protein abundance at 24 hpi were mainly related to processes of primary immune response (Supplementary Fig. 4a, b, Supplementary Data file 3 IV and VI). Of note, a comparison of immortalised and primary fibroblasts revealed similar proteomic profiles (Supplementary Fig. 4c, d, Supplementary Data file 3 VII and VIII). We therefore performed a meta-analysis considering, all -to our knowledge- published studies combining ZIKV infection and mass spectrometry-based analysis of global or proximity labelled proteomes in human cells (Fig. 5a, 12 publications with consistent datasets available, Supplementary Table 4). The resulting dataset (Supplementary Data file 4VI) comprises 10,427 entries with proteins affected by ZIKV of which 2,801 were classified as enzymes (Fig. 5b, Supplementary Data file 4V), including 14 enzymes potentially involved in the production of VOCs (Fig. 5c, Supplementary Table 5, Supplementary Data file 4IX). Among the 2,801 enzymes affected by ZIKV, we observed enrichment of enzymes involved in the response to oxidative stress as well as enzymes acting on alcohols and carbonyls (Supplementary Data file 4 VII and VIII). Of note, there was a highly significant enrichment of enzymes involved in fatty acid derivative metabolic processes (Fig. 5b). Specifically, the alcohol dehydrogenase class-3, 3-hydroxyacyl-CoA dehydrogenase type-2, the mitochondrial acyl-CoA dehydrogenase family, and the mitochondrial very long-chain specific acyl-CoA dehydrogenase, all of which were upregulated upon ZIKV infection. All these enzymes are associated with the oxidative as well as the reductive degradation of lipids and fatty acids (Fig. 5b, c, Supplementary Data file 4), both of which are important for the formation of the VOCs described here (Supplementary Figs. 5 and 6). However, a number of other fatty acid-modifying enzymes must be present, including those responsible for methyl branching as well as reducing enzymes that form alkanes and alkenes, but these have not yet been described in the context of

human cell metabolism and therefore cannot be present or discussed in our analyses. In sum, -omics meta-analyses reveal ZIKV-induced upregulation of lipoprotein and lipid metabolism enzymes that potentially are linked to Virus Induced Mosquito Attractant (VIMA)-VOCs release.

## Discussion

When we investigated the *Ae. aegypti* mosquito behavioural response to the synthetic-related odour of ZIKV infected human cells in invasion and transmission stages of ZIKV infection, we observed definite and striking effects that may plausibly be linked to viral transmission success. Specific VOCs became more abundantly emitted upon virus infection of fibroblast cells. Those were represented by two major groups: (i) oxidized compounds such as alcohols and carbonyls; and (ii) hydrocarbons. It was reported that VOCs attributed to these two groups compose emissions of B lymphoblastoid cells upon infection with three influenza virus subtypes H9N2, H6N2, and H1N1[13]. The emission of alcohols and carbonyls is associated with induced oxidative stress and low pH conditions reported during viral infection of cells[14]. It is likely that the hydrocarbons detected in our study are produced by oxidative fragmentation of lipids[15]. Taking into consideration that ZIKV and other viruses modify different reactions in lipid biosynthesis, it is expected that some virus-specific hydrocarbons could be formed[13,16,17].

Previous studies indicate that vector-borne pathogens like the plasmodial parasites causing malaria and their metabolites can cause an increase in the amount of blood imbibed by the mosquito that could potentially increase not only the odds of imbibing infectious sexual forms but also nutrient gain and enhance the reproductive capacity of the vector[18–20].

Lipoproteins are biochemical assemblies, whose role is to transport lipids in the extracellular fluid. Lipoproteins play a key role in the absorption and transport of dietary lipids and the transport of lipids from the internal organs to peripheral tissues. A secondary function is to transport toxic foreign hydrophobic and amphipathic compounds, such as bacterial endotoxin, from areas of invasion and infection. The proteins encoded by the apoL gene family are found in the cytoplasm, where they may affect the movement of lipids or allow the binding of lipids to organelles[21]. It seems infection enhances the lipoprotein levels. Lipoproteins increase the lipid transport in organelles and fragmentation of lipids, which ultimately might induce the volatome alteration during the ZIKV infection. Accordingly, the volatome profile collected during the phase of high transcriptional activity of apolipoprotein L genes (24 hpi) may indirectly reflect a feeding persistency effect on the vector, which presented as an increased phago-stimulatory behaviour in *Aedes* mosquitoes (Fig. 3a, b). A previous study showed that flaviviruses, including dengue and Zika, can manipulate the host's skin microbiota to produce acetophenone, a mosquito attractant. This occurs by suppressing RELMα, an antimicrobial protein in the skin, which allows acetophenone-producing bacteria to proliferate. The resulting increase in acetophenone levels in infected host enhances their attractiveness to mosquitoes, thereby promoting viral transmission[22]. Our study discovered that the ZIKV not only manipulates the skin microbiota but also directly alters the gene and protein expression in human dermal fibroblasts. These changes in host skin cell metabolism led to an increased release of VOCs that are more attractive to mosquitoes.

Aliphatic volatiles, including methyl-branched aliphatic volatiles, are widely distributed in nature, particularly in microorganisms, fungi, and plants, but insects also emit such volatiles. In humans, branched fatty acids have been detected in various tissues and fluids, including adipose tissue[23] and serum[24]. Nicolaides and others[25,26] found that the meibomian and sebaceous glands of human skin produce branched fatty acids. In vitro and in vivo studies in mouse models confirmed the endogenous synthesis of branched fatty acids. Despite this endogenous synthesis relevant for this study, it can be assumed that the dietary intake of branched fatty acids is the major source of branched fatty acids in the human body. Relatively little is known about the biosynthetic origin of these natural products, and to date there are no studies specifically describing their endogenous formation and individual biosynthetic steps in human fibroblasts. It can be assumed that most of the volatiles detected are metabolites derived from linear and

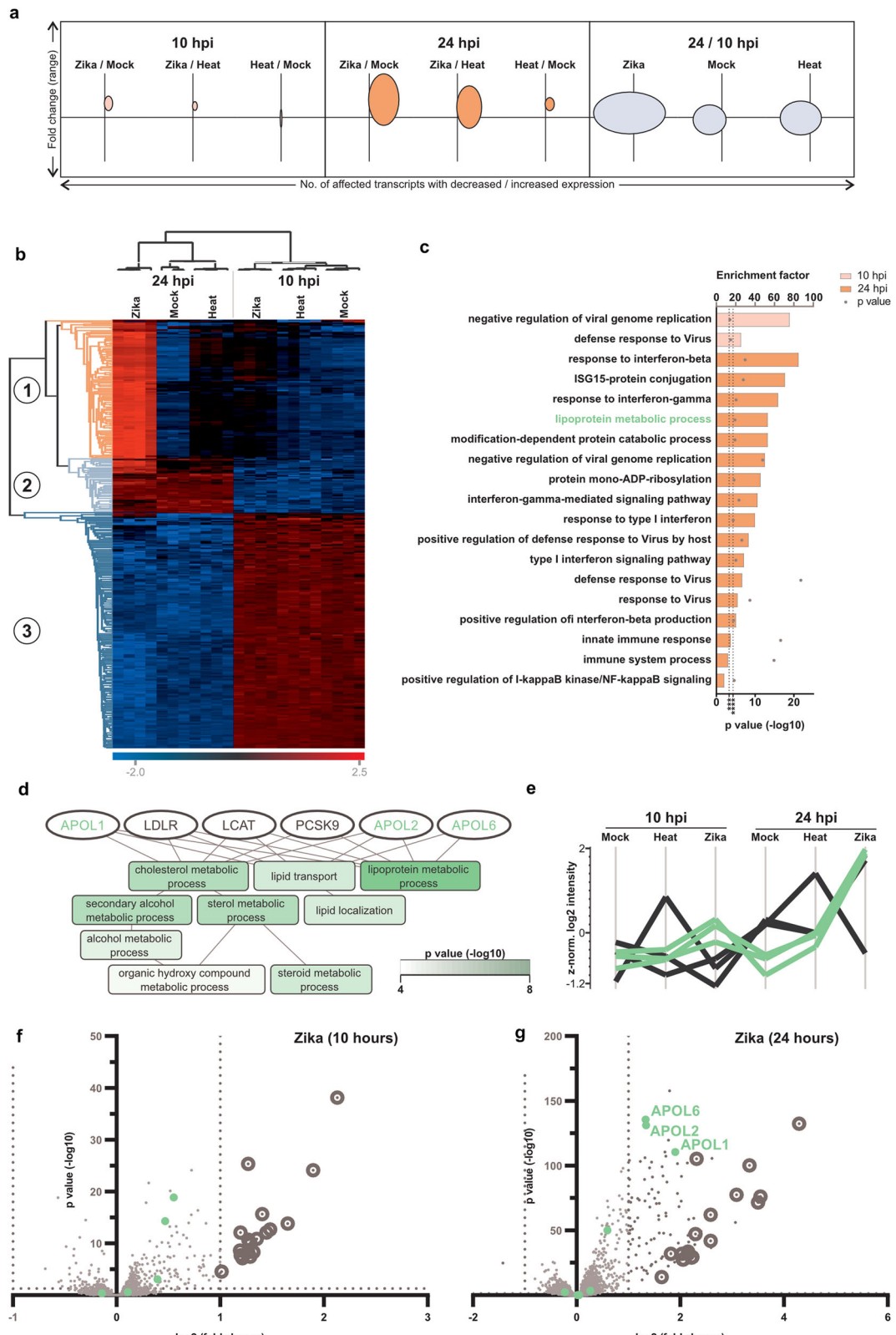

methyl-branched fatty acids (Supplementary Figs. 5 and 6). In order to shed light on their biosynthetic origin, we set out to identify changes in metabolic enzymes that could lead to VOC production upon ZIKV infection. Therefore, we filtered both, the RNAseq and the proteome dataset for all enzymes and, furthermore, for enzymes potentially involved in the processing of fatty acids, lipids, isoprenoids and the production of VOCs,

respectively (Supplementary Table 3 and Supplementary Data file 4I–V). However, although we observed an upregulation of some upstream enzymes potentially involved in VOC synthesis, we could not reveal the complete metabolic pathway altered by ZIKV infection.

The laboratory studies reported here demonstrate that ZIKV infection triggers an enhanced release of VOC attractants from infected human skin

**Fig. 4 | Distinct effects of ZIKV on temporal gene regulation in human skin cells, fibroblasts.** The fibroblast's transcriptome was analysed at 10- and 24-h post-infection (hpi) of cells. **a** Bubble plot of 401 affected transcripts among different conditions. X-axis: number of transcripts with decreased (left of y-axis) and increased (right of y-axis) expression, respectively. Y-axis: range of log2 fold changes (x-axis = zero). The volume of bubbles reflects the effect of treatment. **b** Hierarchical clustering of affected transcripts with three visible clusters. Expression level is reflected by colour (high = red, low = blue). **c** Enrichment of biological processes (Gene Ontology database) among the transcripts with significantly altered expression 10 hpi ($p \geq 0.05$) and 24 hpi ($p \leq 0.05$), independent from control. Dotted lines represent the $p$ values of $5 \times 10^{-4}$ (10 hpi) and $5 \times 10^{-5}$ (24 hpi) respectively, which

were used for the truncation of results. **d** All detected transcripts (ellipses) annotated with *lipoprotein metabolic process* (according to Gene Ontology database) and further biological processes potentially influenced ($p \leq 5 \times 10^{-4}$) by these transcripts. Transcripts labelled green were significantly affected upon infection with ZIKV. Fill colour intensity of biological processes represents the likelihood of a process being affected by these transcripts and their fold changes as observed by RNAseq analysis (according to GO net, $q \leq 5 \times 10^{-4}$). **e** Expression value profile plot of transcripts shown in (**d**). **f, g** Volcano plots of expression levels observed 10 hpi (**f**) and 24 hpi (**g**) with ZIKV, compared to heat-inactivated controls. Green: genes involved in lipoprotein metabolic processes. Circled dots: transcripts already affected 10 hpi. Dotted line at x-axis: $p = 0.05$. Dotted line y-axis: fold change = 2.

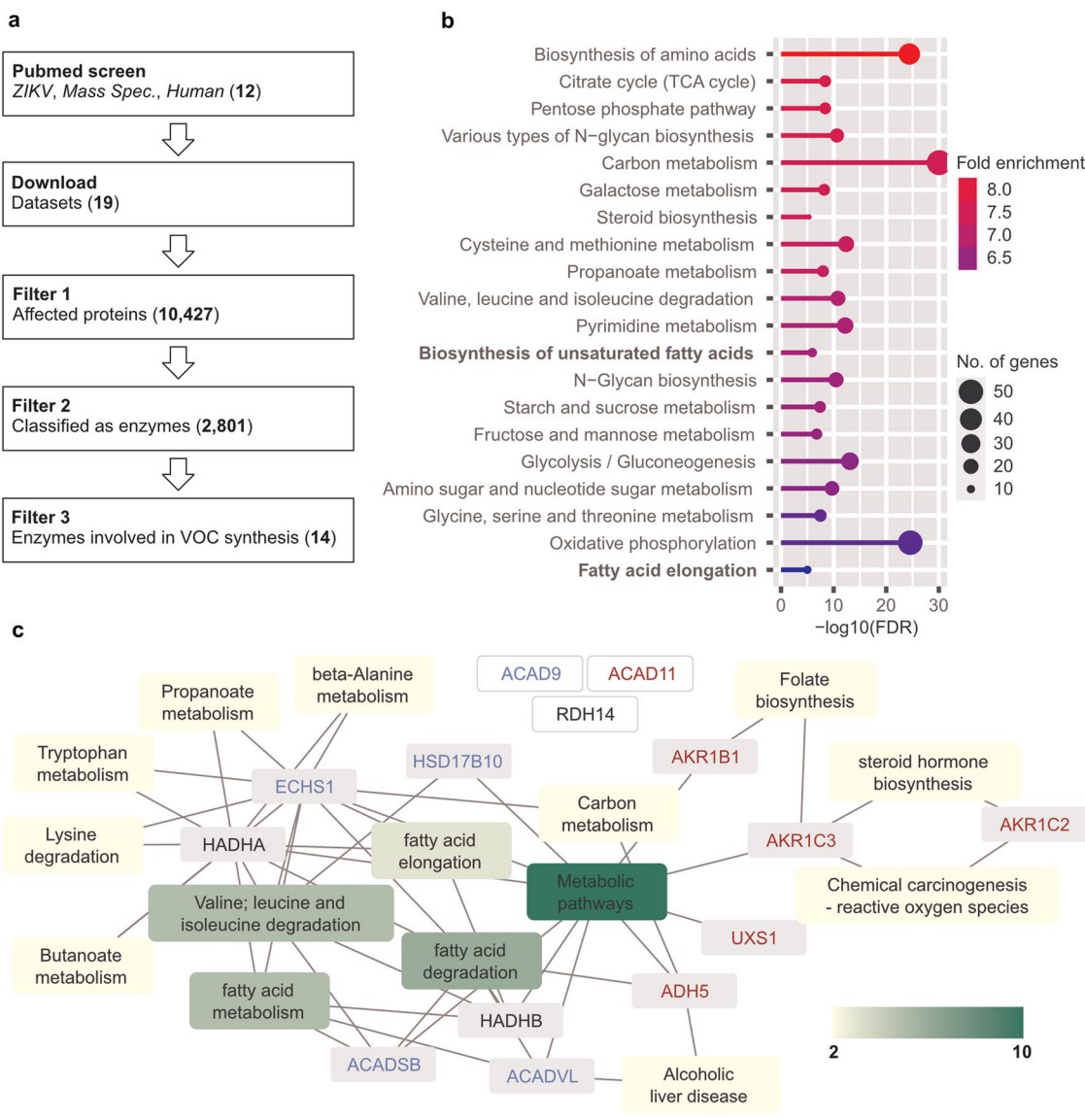

**Fig. 5 | Results of meta-proteome analysis. a** Scheme of the performed meta-proteome analysis comprising 12 studies applying mass spectrometry-based analysis on human cells infected with ZIKV resulting in a dataset with more than 10,000 proteins (entries) affected by ZIKV. **b** Enrichment of KEGG pathways (FDR $\leq 10^{-4}$) among affected proteins classified as enzymes (top 20, ranked by enrichment factors). **c** The meta-proteome analysis was screened for enzymes potentially involved in the synthesis of VOCs, which were subsequently annotated with related KEGG

pathways. Enzymes related to different KEGG pathways are depicted grey boxes or white boxes, if not related to any KEGG pathway. The colour of labelling represents ZIKA-induced changes observed in the meta-proteomics analysis (red: increased expression, blue: reduced expression, black: interaction, see table S5). KEGG pathways related to more than one enzyme are depicted in yellow or green boxes, depending on the number of KEGG pathway relationships.

cells and that these provoke significant alterations to mosquito blood meal-seeking and—obtaining behaviours. Those plausible results indicate an increased likelihood that an infected human host gets bitten by the vector. Moreover, ZIKV odour indirectly stimulates the probing and increase of blood meal size, leading to a greater probability of virus transmission. Transcriptome and proteome meta-analyses confirm that ZIKV infection is critical in that it modulates the expression of lipid metabolic and lipid transport proteins in its human host's cells. These phenomena may represent an evolved means of augmenting its transmission from one human host to the next by manipulation of the vector (Supplementary Fig. 7). There are potentially profound implications of this previously unknown set of arboviral manipulations of its vertebrate host in manners that provoke transmission-enhancing behaviours in its invertebrate host. Among these are exploitation against arboviral transmission success by efficient baiting and lethal trapping as a more effective, sustainable, and environmentally benign approach to the very difficult problem of mosquito control.

## Methods

### Ethics

Human blood (type O) was provided in citrate-phosphate-dextrose-adenine anti-coagulant/preservative, and serum (type AB) was obtained from the Blood Transfusion Service at Karolinska Hospital, Solna, Sweden in accordance with the Declaration of Helsinki and approved by the Ethical Review Board in Stockholm (850-32). The Swedish Work Environment Authority, Stockholm, Sweden (Dnr SU FV-2.10.2-2905-13/31-01-2017) approved the class 3 biological agent laboratory and practices, including insectary design and equipment to work with infected mosquitoes. The ordinances are mainly based on the EU directive 2000/54/EC on the protection of workers from risks related to exposure to biological agents at work. The patients or their parents provided informed consent granting the use of their samples and associated data and images in this publication according to the requirements set forth by the Institutional Review Board at Erasmus University Medical Center (protocol METC-2012387), the National Institute of Diabetes and Digestive and Kidney Diseases/National Institute of Arthritis, Musculoskeletal and Skin Diseases Institutional Review Board, or the Hacettepe University Ethics Committee. Cultures of fibroblasts were prepared from the foreskin as described previously[27]. All patients gave written confirmed consent to participate in the study. The procedure to use foreskin from anonymised patients was approved by the Ethical Committee of Hannover Medical School, Hannover, Germany.

### ZIKV infection of human fibroblasts

Human dermal fibroblasts (hTert-immortalized[28]), a kind gift from Prof. Dusan Bogunovic, were maintained in growth media containing Dulbecco's Modified Eagle Medium (Sigma Aldrich) supplemented with 1% non-essential amino acids (Gibco, Thermo Fisher Scientific), 100 U/mL penicillin & 100 µg/mL streptomycin (Gibco, Thermo Fisher Scientific) and 10% fetal bovine serum (FBS, Cytiva) at 37 °C in 5% $CO_2$. ZIKV (ZIKV; isolate PE243)[29] was propagated in C6/36 *Aedes albopictus* cells, as previously described[29]. Human dermal fibroblasts ($4 \times 10^6$ cells) were seeded in 175 cm² cell culture flasks one day before infection. At the time of infection, ZIKV was added to the cells in growth media containing 2% FBS at a multiplicity of infection (MOI) of 0·1, or mock treated for uninfected control cells and incubated at 37 °C in 5% $CO_2$. Two hours post-infection (hpi), the inoculation material was removed, and the cells were washed twice with phosphate-buffered saline (PBS). New growth media containing 2% FBS was added to the cells and the infection proceeded for a total of 10 hpi (invasion stage) and 24 hpi (transmission stage). Six hours prior to VOC sampling, the cells were detached using 0.05% Trypsin-EDTA (Gibco, Thermo Fisher Scientific) and washed once with PBS. New media was added to the cells and $6 \times 10^6$ cells in a total volume of 6 ml were transferred into a 7 cm³ glass bottle. The cells were kept in suspension by gentle agitation until sampling. In addition to the ZIKV infected cells at 10 hpi ($n = 6$), at 24 hpi ($n = 6$) and uninfected control cells ($n = 6$), cell culture media ($n = 6$) and

ZIKV diluted in cell culture media ($n = 6$) were also incubated in a total volume of 6 mL in 7 cm³ glass bottles at 37 °C six h prior to VOC sampling.

### Sampling and identification of VOCs

Odour samplings were carried out by solid phase microextraction (SPME) technique. Prior to sampling, the polydimethylsiloxane/divinylbenzene-coated SPME fiber was purified for 1 min at 230°C in a gas chromatograph (GC) (Hewlett Packard 6890 N, Agilent Technologies Inc., Santa Clara, CA, USA) injector. Afterwards, the fiber was inserted into the 7 cm³ glass bottle and exposed to the headspace. Volatiles were collected for 60 min and afterwards transported to the analytical laboratory. Analyses were carried out by GC coupled to an HP 5973 mass spectrometer (MS) (Hewlett Packard 6890 N, Agilent Technologies Inc., Santa Clara, CA, USA). The volatiles from SPME fiber were desorbed in the injector (spitless mode, 1 min, 230 °C). Helium was used as the carrier gas with a constant flow of 1 mL min⁻¹. The GC was equipped with a DB-5 silica capillary column (30 m length, 0·25 mm ID, 0.25 µm film thickness). The GC oven temperature was held isothermal at 40 °C for 2 min, afterwards increased by 4 °C min⁻¹ up to 200 °C and then increased by 10 °C min⁻¹ up to 280 °C. Electron ionization mass spectra were determined at 70 eV with the ion source at 150 °C. Compounds were identified by comparison of their retention indexes and mass spectra with those available from NIST mass spectral database, version 2.0 (National Institute of Standards and Technology, USA) and from available publications as well as by comparing retention indexes and mass spectral data of natural products with those of authentic reference standards. Chromatographic profiles of the volatiles were compared and the compounds which occurred in significantly large amounts in the infected cell samples compared to those of blank samples were further analysed.

Invasion and transmission blends comprised of synthetic, olfactory active compounds representing emissions from fibroblasts infected with ZIKV at invasive and transmission stages, respectively have been formulated, respectively. 4-Methyl-heptane, 2,4-dimethyl-heptane, sulcatone, decanal, and dodecanal at loadings of 0.7, 6, 0.2, 150, and 1100 µg mL⁻¹, respectively in paraffin oil, composed the invasion blend, while the transmission blend comprised 2-methyl-1-pentene, sulcatone, decanal, and dodecanal at loadings of 0.1, 0.5, 400, and 3000 µg mL⁻¹, respectively. The loading amounts of synthetic odorants were selected to emit volatiles at ten times larger amounts but keeping a similar ratio to those of fibroblasts infected with ZIKV at invasive and transmission stages.

All single compounds (2-methyl-1-pentene, 4-methyl-heptane, 2,4-dimethyl-1-heptene, sulcatone, decanal, and dodecanal) were purchased from Sigma Aldrich (St. Louis, Missouri, USA).

### Electroantennographic tests of selected synthetic VOCs

Six commercially available synthetic VOCs, i.e., 2-methyl-1-pentene, 4-methyl-heptane, 2,4-dimethyl-1-heptene, sulcatone, decanal, and dodecanal analogous to the compounds whose amount increased in emissions of virus infected versus not infected cell cultures were tested for electroantennographic responses using mated five to seven days old *Ae. aegypti* females. Females were allowed to feed on sucrose solution before the testing. *Ae. aegypti* used in EAG analyses were not chilled prior to the test. The head was removed with scissors, and the tip of the antenna was cut off. Glass capillary electrodes were filled with physiological solution (NaCl 0.9%, Ilsanta, Vilnius, Lithuania). The reference electrode was inserted into the hemocoel of a female's head and the recording electrode was joined with the cut tip of the antenna and connected to a high-impedance DC amplifier (IDAC-4, Ockenfels Syntech GmbH, Buchenbach, Germany) with automatic baseline drift compensation. Activated charcoal-filtered and humidified air flow at a rate of 1 L min⁻¹ passed over the antenna through a glass tube (ø 6 mm) positioned 0.5 cm from the insect preparation. Pulse airflow with odorant was pushed into the constant airflow for 0.5 second (sec). Synthetic VOCs were diluted with ether and 10 µL of solutions at a concentration of 1 mg mL⁻¹ were applied on a filter paper for EAG screening. The test materials were presented to a female antenna at approximately 1 min intervals and the order of presentation was random. Twenty-one

replications of EAG registration were performed. Electroantennographic registrations were recorded at room temperature (24 ± 2°C). The antennal signals were recorded, stored, and analysed using GcEad V. 4·4 software (Synthech, Ockenfels Syntech GmbH, Buchenbach, Germany).

## Mosquito rearing and blood feeding

In this study, mosquitoes were sourced from two laboratory colonies. The *Aedes aegypti* mosquitoes were obtained from a colony maintained at the Liverpool School of Tropical Medicine in the UK. The *Anopheles gambiae* sensu lato (Keele line) mosquitoes were produced by balanced interbreeding of four distinct *An. gambiae* s.s. strains: KIL from Marangu, Tanzania; G3 from MacCarthy Island, Gambia; ZAN U from Zanzibar; and Ifakara from Njage, Tanzania. This Keele colony was established in Glasgow in 2005. Both mosquito colonies have been sustained through membrane feeding with human blood in the University's insectaries[30].

 Larvae were reared under standard insectary conditions (27 ± 1 °C, 70% humidity, 12 h light: 12 h dark cycle) and fed on TetraMin fish flakes (Tetra ltd., Germany). Pupae were transferred into holding cages for emergence. Emerged adults were fed *ad libitum* on 5% glucose solution, supplemented with 0.05% (w/v) 4-aminobenzoic acid (PABA, Sigma-Aldrich), through soaked filters on top of the 2 mL tubes and with soaked filter pads inside cages. Red blood cells (RBCs) were washed with Roswell Park Memorial Institute (RPMI) medium and stored in RPMI at 50% haematocrit at 4 °C until use. To run feeding experiments, RBCs stored in RPMI were centrifuged at $2500 \times g$ for 5 min followed by replacement of the medium with AB serum for a final haematocrit of 40%. All experiments were, unless otherwise stated, conducted on five to seven days post-emergence female mosquitoes maintained in separated cages (approximately 30 individuals per cohort) and fed RBCs either with or without impregnated ZIKV odours (invasion and transmission blend) for 800 s. All experiments were performed in triplicates.

## Mosquito fitness

Three groups of mated five to seven days old *Ae. aegypti* female mosquitoes ($n \approx 30$ in each replicate) were allowed to feed for ten min on RBCs with or without nearby located ZIKV blends (invasion and transmission) and were sorted into individual holding tubes with sugar pads. At three days post-ingestion (dpi), the amount of RBCs imbibed was estimated using the hematin excretion assay[31]. Female mosquitoes were transferred to tubes containing 1 cm of water at the bottom for egg-laying. On day 5 dpi, the number of eggs laid per mosquito (fecundity) was counted. Mosquito holding tubes were examined daily until the death of each mosquito (survival). The body size of mosquitoes was estimated post-mortem by measuring wing length under a stereo-microscope (Nikon ND2, Tokyo, Japan) with a measuring eyepiece[32,33]. Mosquitoes were offered a 5% glucose solution containing 0·05% PABA using fresh soaked sugar cotton daily at the top of their holding cages. All experiments were performed in triplicates.

## Dual choice assay

Initial experiments were performed using a Y-tube olfactometer (length of the central cylinder and two arms: 25/15/15 cm respectively, inner diameter: 5 cm; Fig. 1f). For each experiment, 30 mated five to seven days old female mosquitoes were individually placed in a release chamber and flown one by one. Females flew upwind in the central cylinder (airflow 3 L min⁻¹) and entered one of the two arms equipped with a trapping chamber located at the upwind end of the arm. The trapping chamber was connected to a stimulus delivery tube bearing 60 µL paraffin oil with or without stimulus. Two sets of experiments were conducted. In the first experimental set, the stimulus delivery tubes in the left and right arms were loaded with solvent with or without a blend of ZIKV's VOCs (transmission or invasion stage) for *An. gambiae* s.l., while in the second stimulus delivery tubes were loaded with solvent with or without a blend of ZIKV's VOCs (transmission or invasion stage) for *Ae. aegypti*. To control for possible spatial effects, the location of treatment at each olfactometer

arm was switched every 30 min ( ~ 15 mosquitoes flew one by one every 30 min). Each replacement (holder rotation with fresh blend's solvent) was counted as an experimental replicate (experimental block). Mosquitoes reaching any of the trapping holder chambers were considered to have made a choice. Each experiment was repeated three times with a total of 90 mosquitoes per treatment.

## Bioactivity tests of synthetic VOCs in the wind tunnel

The locomotor and flight activity of mosquitoes were assayed using the 3D Noldus videography software (Noldus, Netherlands). Thirty female five to seven days-old mosquitoes, which had not received a blood meal, were transferred to the experimental cage 14–16 h before testing. A water-soaked filter paper was placed in the cage as a meal for the mosquitoes to prevent the desiccation of the mosquito. Individual *Ae. aegypti* females were transferred into the Noldus EthoVision wind tunnel (1.6 m), to analyse locomotion and flight activity in the wind tunnel upwind towards the mesh screen and the data was collected after the whole experiment and analysed. Mosquitoes that were not within the field of view of both cameras after release were recorded as 'no response'. Recordings were stopped after one minute of recording initiation. The wind tunnel's chamber where the mosquitoes are released is an arena. The arena is the internal area of the chamber, where the observation is taking place. The arena is thus divided into two zones. Zone one is close to the release area of the mosquitoes into the flight chamber and zone two is close to the plume area where the 60 mL paraffin oil with or without stimulus are placed. Out of the total number of mosquitoes that entered the arena, the ones that proceeded to zone two and stayed in zone two for more than 15 s are regarded as positive reactions to the blends. Mosquito response was categorized as either 'no response', positive response or negative response, any mosquito out of the arena is regarded as no response. Mosquitoes in the arena are regarded as responsive and the mosquitoes that proceeded to zone two are a positive response while those that remained in zone one are a negative response to the experiment. A new set of mosquitoes were used for each bioassay and three technical replicates were made in every biological replicate of the experiment. Surgical gloves were worn to avoid contamination of the experimental equipment and to avoid any trace of human smell.

## Feeding proportion experiments

Thirty females *Ae. aegypti* were isolated in three separate cages (8 cm internal diameter × 10 cm in height) covered with netting and fed on 1 mL of control RBCs (without nearby located blends), and feeder containing RBCs with nearby located ZIKV related-blends, invasion or transmission, respectively. Females five to six post-emergence days were used for the experiment. The female mosquitoes used for the experiment have been mated since post-emergence. The glucose meal in the cage was replaced by distilled water to avoid desiccation for 24 h before the experiment was carried out. Three replications were done for the control, invasion, and transmission treatments respectively. The blood meal (RBCs) was placed over the net with the help of a glass feeder, and the blood meal uptake initiation, feeding proportion and latency were observed for 800 s. Accumulated blood feeding proportion (%) of mosquitoes was monitored per ten sec at the first minute of the experiment and then continued per minute on feeders with/without nearby ZIKV related-odour at the invasion or transmission stage (invasion blend or transmission blend) until 800 s. Within this period any mosquitoes that have fully fed from the feeder and disengaged were also counted as blood-fed mosquitoes. This procedure was repeated three times for each of the three treatments: meaning for control there was paraffin oil placed nearby the feeder but as for the invasion and transmission experiments, the VOC blends were placed nearby the feeder to observe if the mosquitoes would react to the smell from the blend and thus feed more on the blood or not. Each group was allowed to feed on its separated membrane feeder for 800 sec, the average time for mosquito vector engorgement[20,34]. For each group, the number of fully-fed mosquitoes compared to unfed controls was immediately recorded.

## Transcriptomics experiments

Human dermal fibroblasts ($0.275 \times 10^6$ cells) were seeded in 6-well cell culture plates one day prior to infection in growth media supplemented with 10% FBS. The cells were infected with ZIKV or heat-inactivated ZIKV (70 °C, 15 min) at a MOI of 0.1 diluted in growth media containing 2% FBS, or mock treated, for uninfected control cells, treated cells with heat-inactivated virus also serves as uninfected control. The cells were incubated at 37 °C and 5% $CO_2$ for 2 h before the inoculation material was removed and the cells were washed twice with PBS. New growth media containing 2% FBS was added to the cells and the incubation continued for a total duration of 10 hpi (invasion stage) or 24 hpi (transmission stage). At the time of cell harvest, media was removed, and the cells were washed twice with PBS and then lysed by the addition of lysis buffer (Macherey-Nagel, Nucleospin Viral RNA kit). The samples were stored at -80 °C until RNA extraction according to the instructions from the manufacturer. For each biological replicate, three wells with cells were pooled, and a total of four replicates per condition (three replicates for mock 24 hpi) were subjected to bulk RNA sequencing (RNAseq) after exclusion of viral contamination as performed by qPCR analysis (data not shown).

Bulk RNAseq was performed with Illumina NovaSeq6000 S4 (Novogene) at the Research Core Unit Transcriptomics of the Hannover Medical School. Raw data processing was done by applying nfcore/rnaseq (Nextflow) and the DESeq2 algorithm[35]. Genome reference and annotation data were taken from GENCODE.org (Homo sapiens: GRCh38·p13; release 34). Differential gene expression analyses were performed separately applying the DESeq2 algorithm to identify significantly altered expression across all conditions 10 hpi and 24 hpi, respectively, and by the course of time within the conditions. Cellular compartments, biological processes and molecular function of transcripts (GO terms: GOCC, GOBP, GOMF) were annotated based on Ensembl Gene IDs using the R/Bioconductor package biomaRt[36]. Downstream analysis was performed using the *Perseus* software (V.1·6·15·0, Max Planck Institute of Biochemistry[37]). The 39,428 identified transcripts were filtered for transcripts reaching the defined expression value threshold of 150 in 65% of replicates in at least one condition followed by log2 normalization and replacement of missing expression values by random values drawn from a normal distribution (width: 0.3; down-shift: 1.8; separately for each column). The resulting list of 11,535 transcripts was subjected to principal component analysis (Supplementary Fig. 3) and visualized as volcano plots (Fig. 4f). Significant change of expression was defined by a fold change ≥ 2 and q value ≤ 0.05, both derived from differential gene expression analyses, resulting in 401 transcripts annotated as "affected". Log2 fold changes of affected transcripts and the number of affected transcripts across conditions were visualized as bubble plots while z-normalized expression value medians of all conditions were used for hierarchical clustering (Fig. 4a, b) followed by defining three row clusters. Enrichment of GO terms among transcripts with significantly altered expression 10 hpi and 24 hpi with ZIKV, respectively and independent from control, compared to the list of 11,535 transcripts was performed by Fisher exact tests with $p$ values of $5 \times 10^{-4}$ and $5 \times 10^{-5}$, respectively, used for truncation of results (Fig. 4c). To screen for biological processes related to subsets of chosen transcripts we applied the web-based tool *GOnet*[38] using the transcript list with mean fold changes 24 hpi of ZIKV vs. mock and ZIKV vs. heat-inactivated as the input list, the list of 11,535 noise reduced transcripts as the background list and the q value threshold set to $1 \times 10^{-4}$ (Fig. 4d). For visualization of profile plots median fold changes of all conditions were calculated and transformed by z-normalization (Fig. 4e).

## Proteome Analysis

For the comparison of ZIKV-infected to Mock-infected immortalized fibroblasts, cells were cultured and infected at the Department of Clinical Microbiology (Umeå) as described above (Transcriptomics experiments). For the comparison of two primary cell lines (F02-09, F10-09), a kind gift from Prof. Thomas Werfel, to the immortalized cell line (hTert), all cells were cultured at the Research Center for Emerging Infections and Zoonoses (Hannover) in Dulbecco's modified Eagle's medium (DMEM, Gibco,

Thermo Fisher) supplemented with 10% fetal bovine serum (FBS Advanced, Capricorn Scientific), 1% non-essential amino acids, 2 mM L-glutamine, 100 U/ml penicillin and 100 μg/ml streptomycin (all Thermo Fisher). A commercially available HeLa- (Thermo Fisher) and a self-made HEK-cell-derived protein digest standard were used as references. Of each condition/cell line, 4 technical replicates were provided for sample processing and analysis.

Cells were lysed with RIPA buffer on ice for 30 min. Cellular debris were removed by centrifugation for 10 min at 12,000 g and 4 °C and transferring the supernatant to a new reaction tube. 20 μg of protein were taken from the lysate for precipitation overnight using ice cold acetone (final concentration of 80%). Proteins were spun down at 11,000 g for 15 min at 4 °C. Digest of proteins was performed using Trypsin Gold (Promega) following the manufacturer's instructions. Samples were purified using StageTips following the protocol of Rappsilber et al.[39]. Purified peptides were dried in a SpeedVac Vacuum Concentrator and resuspended in MS buffer (0.1% formic acid, 5% acetonitrile in water (MS grade)). Samples were analysed via liquid chromatography–mass spectrometry (LC-MS) using a Vanquish Neo nanoflow UHPLC System coupled to an Orbitrap Eclipse mass spectrometer (Thermo Scientific™). Of each sample, 500 ng peptides were separated on a reversed-phase nanoViper™ PepMap™ separating column (150 mm length, 75 μm inner diameter, and 2 μm C18 particle size (Thermo Scientific™)) using buffer A (80% ACN, 0.1% FA) and B (80% ACN, 0.1% FA). Peptides were eluted at a flow rate of 250 nL/min at 40 °C by a gradient starting with Buffer B increased from 6% to 25% within 60 min and further increased to 90% within 10 min and held at 90% for an additional 8 min. Ionisation was achieved by a Nano Spray Flex Ion Source and stainless steel emitters (40 mm, OD 1/32) at 1,900 V. Precursor scans were performed in the Orbitrap mass analyser with an m/z range set to 375–1500, at a resolution of 120,000, and stored in profile mode. The twenty most intense precursors with intensities above 2000 counts were fragmented in the linear ion trap by higher-energy collisional dissociation with collision energy set to 30% and dynamic exclusion set to 45 s. Fragment ion scans were performed in the Ion Trap mass analyser with AGC set to Standard, Injection Time Mode set to Auto, mass range set to Normal and spectra stored in centroid mode.

MS raw data were processed with MaxQuant software (V. 2·4·14·0) for the identification and quantification of proteins using preconfigured settings and, additionally, selecting label-free quantification via LFQ (Min. ratio count set to 1) and IBAQ as well as writing mzTab tables. MS spectra were searched against the UniProt database of human (ID UP000005640, reviewed, database downloaded on 04/03/2024) and ZIKV proteins (Uniprot entry A0A1Z2X283).

The results were analysed with the *Perseus* software (V.1·6·15·0, Max Planck Institute of Biochemistry[37]) based on LFQ values and applying preconfigured settings. From the resulting protein list, contaminants, proteins only identified by site modifications, proteins identified in the decoy database and proteins identified by only a single peptide were removed as well as bad replicates with less than 1000 quantified proteins. Further analyses were restricted to proteins with at least 2 valid values per group. Missing values were replaced by low numbers according to the normal distribution of the dataset. Groups were compared by pairwise $t$-tests. For Hierarchical Clustering analysis, intensities were further transformed to z-Scores. Enrichment analysis was performed using the web-based tool *Metascape*[40].

## Meta-Proteome-Analysis

PubMed database (https://pubmed.ncbi.nlm.nih.gov/; 20.02.2023) was screened for studies (no reviews, no meta studies) applying mass spectrometry-based shotgun proteomes on human cells infected with ZIKV (search for *((Zika[Title/Abstract]) AND (Proteomics[Title/Abstract])) AND (Human[Title/Abstract])* and *((Zika[Title/Abstract]) AND (Proteomic[Title/Abstract])) AND (Human[Title/Abstract]))*. Available and comprehensible datasets comprising mass spectrometry data were downloaded, filtered for significantly affected or enriched proteins and combined into one dataset

(Supplementary Data file 4 VI). Protein names were screened for strings listed in Supplementary Data file 4 II and classified as enzymes based on the strings "ase" or "enzyme" or based on the list of enzymes listed in Supplementary Data file 4 I. Enrichment analysis among proteins classified as enzymes was performed applying *Shiny GO* (V 0.77, http://bioinformatics.sdstate.edu/go/; 23.02.2023) applying standard settings with no background uploaded (Supplementary Data file 4 VII) or FDR set to $10^{-4}$ (Fig. 5b). Annotation of enzymes classified as candidates for VOC synthesis was performed using the web-based tool KEGG Mapper (https://www.genome.jp/kegg/mapper/) using gene symbols as input. Annotated enzymes were processed and loaded into Cytoscape software (version 3·9·1) to generate a network graph.

### Biosynthetic considerations on the origin of VOCs found in fibroblasts

Since virtually almost nothing is known about the biosynthetic origin of methyl-branched alkanes in human cells, especially human fibroblasts, only basic considerations related to this topic can be given in the context of this work, based on known metabolic pathways. These considerations support our meta-proteomic analyses. The majority of the detected volatiles represent metabolites derived from linear and methyl-branched fatty acids. Their existence in various human tissues and fluids, including adipose tissue (17) and serum (18), has been clearly demonstrated. Biosynthetic details of the precursors and the enzymes involved are poorly elucidated, and this is especially true for the late-stage reductions to alkanes and alkenes.

Methyl branching occurs in fatty acid biosynthesis by several routes[41-43]. The methyl groups in the central part or at the carboxylate terminus can either originate from methyl malonate or be incorporated by a radical SAM-mediated process (SAM = S-adenosylmethionine). The methyl malonate version is applicable to the biosynthesis of the precursor fatty acid of 2,4,6-trimethyldecane, 2,7-dimethyl-1-octanol, and 2,2,4,4-tetramethyloctane (Supplementary Fig. 5), whereas the radical SAM version is basically applicable to all methyl-branched alkanes and alkenes. This is because the methylmalonyl route only allows incorporation at every other position of the linear carbon skeleton, whereas SAM-mediated methyl transfer can in principle address any carbon atom. Third, the branches near the reducing end of the precursor often originate from the branched aliphatic amino acids valine, leucine, and isoleucine, which serve as the starting building blocks of fatty acid biosynthesis[44-46].

The reductive removal of the carboxylate required to form alkanes or alkenes can occur in various fashion[41-43]. On the route to alcohols, as in the case of 2,7-dimethyl-1-octanol, the intermediate aldehyde (similar to decanal) is probably further reduced by alcohol dehydrogenase. At this oxidation state, the reduction of SCoA fatty acid esters commonly halts. Alkanes are formed either by a decarbonylation step at the aldehyde level, catalysed in higher organisms by aldehyde deformylating oxygenase or aldehyde decarbonylase. These enzymes occur in bacteria and cyanobacteria but have not yet been described in mammals. Alternatively, alkanes are produced biologically by decarboxylation of fatty acids catalysed by FAD-dependent fatty acid photodecarboxylase. Finally, fatty acid-derived alkenes are formed by decarboxylation, for which cytochrome P450-dependent fatty acid decarboxylase is responsible, first found in microalgae. At present, it is not clear by which biosynthetic pathway branched-chain fatty acids are degraded to alkanes and alkenes in fibroblasts, mainly because no data on fatty acid metabolism in fibroblasts have been published.

Based on these facts we propose a fatty acid metabolism that yields VOCs reported here. The possible biosynthetic routes are visualized using a colour code for the carbon atoms (Supplementary Fig. 6a–e). This code refers to the biosynthetic precursor building blocks. Biosynthetically, long-chain fatty acids precursors may be shortened by ß-oxidation first before reductive steps and removal of the carboxylate terminus finalise the generation of the VOCs (Supplementary Fig. 5).

Finally, 6-methyl-5-heptene-2-one (Sulcatone) and 6-methyl-6-heptene-2-one are among others known mosquito attractants[47]. In humans, it is likely derived from geranyl-pyrophosphate which is part of the mevalonate pathway. This pathway initiates the biosynthesis of steroids and is responsible for the formation of cholesterol. Oxidative degradation by a dioxygenase leads to an intermediate that collapses (enzymatically or chemically) to Sulcatone from where 6-methylhept-6-ene-2-one is formed by double bond isomerization (Supplementary Fig. 5b). Our consideration resulted in a list of 15 enzymes potentially involved in the processing of fatty acids, lipids, isoprenoids and the production of VOCs, respectively (Supplementary Table 3, Supplementary Data file 4).

### Statistics and reproducibility

GLMM statistical modelling was used to corroborate the validity of results based on the whole data set by including the effect of replications (experimental blocks), including weighting for multiple replications. In all analyses, the effect of the main experimental effects (e.g., treatment) was investigated while controlling for variation in experimental replication (random variable). For all results, the significance of all explanatory effects was evaluated by using the likelihood ratio test (LRT). Analyses were performed using R statistical software.

In all analyses, ZIKV blends from invasion and transmission stages were investigated as the primary effect of interest. Generalized Linear Mixed Models (GLMM, R statistical software V·4.2.1)[48] assuming a binomial distribution was used to test the effect of ZIKV blend on the binary response variable of dual choice in the attraction, feeding rate assays (Logistic regression models, absent or present; lme4 package, glmer, R, V·4.2.1). Logistic regression is a powerful statistical method for binomial outcome (take the value 0 or 1) with one or more explanatory variables. In this study, we included at least two variables: 1-Treatment (main effect) and 2-Experimental blocks (random effect). In all analyses, treatment (e.g., blood with/without ZIKV odour in two virus stages) was investigated as the main effect of interest[49]. A similar approach was used to test for variation in mosquito fecundity between different experimental treatments. Given the highly over-dispersed nature of parasite abundance data, negative binomial distribution was assumed in these GLMMs (glmmADMB, nlme package, R, V·3.1.1). For blood meal size, fecundity and survival, a backwards elimination approach was used to test for the significance of all fixed effects (ZIKV blend treatment, body size) and their interactions, while controlling for random variation within each replicate. Mosquito body size was fitted as an additional fixed explanatory variable in all cases due to considerable influence in variation of mosquito feeding and fitness parameters[50]. Time series analyses (e.g., survival analysis) were conducted using the Cox proportional hazards model in the R statistical software (V·3.1.1) to assess whether mosquito longevity (days until death) varied between experimental treatments, and also the same model used for the studying the accumulated blood feeding proportion of mosquitoes during 800 s. In these analyses, a frailty function was used to integrate the random effect of replicates into the Cox model with ZIKV blends treatment. From this maximal model, non-significant terms were sequentially removed through backward elimination to reach the minimal statistically significant model[51]. Analysis was restricted to estimating variation in mosquito survival up to natural death, sequentially. Experiment replication was treated as a random variable in all statistical mixed models. All data conformed to the assumptions of the test (normality and error homogeneity). In all mixed models, a maximal model was built that included fixed effects plus the random effects of the experimental replicates.

### Data availability

The relevant data are available in the manuscript and the supplementary information. The data discussed in this publication have been deposited in NCBI's Gene Expression Omnibus and are accessible through GEO Series accession number GSE242165. The raw data are presented in Extended data file1-3.

https://doi.org/10.1038/s42003-025-07543-9 **Article**

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

## Acknowledgements

We thank the Swedish Research Council (Vetenskapsrådet) for funding to S.N.E., (VR/2017-01229) and (VR/2017-05543 UFNW) network. We highly appreciate the support of Molecular Attraction AB, Dr Lech Ignatowicz, and Johan Paleovrachas, as well as Matt Mullenweg (Audrey Capital LLC.). This work was supported by the Knut and Alice Wallenberg Foundation and the German Federal Ministry of Education and Research together with the Ministry of Science and Culture of Lower Saxony through the Professorinnen Programm III to G.G. We thank the Research Core Unit Genomics (RCUG) at Hannover Medical School for providing detailed advice on the experimental design of the bulk RNAseq study, coordinating the implementation, and performing the quality control and all raw data processing steps. This study was also supported by Lithuanian state grant through Nature Research Centre, Vilnius, Lithuania available to R.M., S.R., V.A., and R.B. program 2 Climate and Ecosystems.

## Author contributions

S.N.E. and G.G. conceived the core of the study. S.N.E. and R.M. conceived the volatomic section of the study. The dual choice assay was designed and carried out by S.N.E. and R.M. and mosquito behavioural 3D tracking was performed by MMA. S.N.E. designed, and MMA performed the feeding assay. S.N.E. designed, and J.B. performed the mosquito fitness assay. S.N.E. designed and performed the SPME collection, R.M. performed the quantitative and qualitative GC-MS analyses, M.R. and M.H. analysed the statistical models to define the significant volatome. SNE and RM formulated synthetic blend and designed the release of significant selected volatiles for bioassays. SNE designed and J.B. performed the mosquito fitness experiments. A.S.L. performed ZIKV culturing and TH conducted the additional experiments on primary fibroblast cell lines. All statistical analyses were performed by M.R. and M.H. S.R., V.A., and R.B. designed the EAG and performed the analyses. G.G., A.S.L., and K.C. conceived the temporal transcriptome section of the study and prepared RNA for Illumina sequencing. KC performed bioinformatics analyses of the sequencing and proteomics data. A.K. carried out the biosynthetic analysis. S.N.E. designed and M.H. created the artwork. R.M., M.R., M.H., A.S.L., L.I., K.C., G.G., A.K., and S.N.E. wrote the manuscript. All authors proofread and commented on the final draft of the manuscript.

## Funding

## Competing interests

We declare no competing interests.
