## [Transparent Peer Review file · Communications Biology]

Zika virus modulates human fibroblasts to enhance transmission success in a controlled lab-setting

Corresponding Author: Professor S. Noushin Emami

This manuscript has been previously reviewed at another journal. This document only contains information relating to versions considered at Communications Biology.

Version 0:

Reviewer comments:

Reviewer #1

(Remarks to the Author)

This paper entitled "Zika virus modulates human skin cells to enhance transmission success" describes measurements of the VOCs in ZIKV-infected dermal fibroblasts in tissue culture. The authors found that upon ZIKV infection, at two timepoints (10h and 24h post-infection), various VOCs were differentially produced relative to uninfected cells. When using synthetic versions of these compounds, it was found that they altered mosquito behaviour (attraction and subsequent feeding). The authors propose that the virus is doing this to improve transmission success.

The scientific methods of the paper appear sound. The conclusions are mostly justified however I believe some additional commentary about the limitations of the study are necessary, as listed below.

1. The authors state in their titles that the virus mediates human skin cells, however they just looked in in vitro models, not in actual human skin. There was also no direct evidence of enhanced transmission success in these experiments, just surrogates of transmission. Therefore I think this title could be toned down a bit to reflect this.
2. In the same vein, there has been work demonstrating the impact of the microbiota of the skin being involved with altering mosquito attraction (e.g. <https://doi.org/10.1016/j.cell.2022.05.016>). Perhaps some discussion on this, placing the results here in the context of the microbiota on the skin, would be beneficial to the readers.

Reviewer #2

(Remarks to the Author)

This manuscript suggests that ZIKV infection is modulated by VOCs in infected dermal fibroblasts and alters the emission of VOCs at the invasion and transmission stage. The authors used a human dermal fibroblast cell line to examine the VOC profile during ZIKV infection and analyzed related gene expression using RNA sequencing and proteomics. While the overall topic is interesting, weaknesses in experimental design and interpretation limit the impact of the results. Furthermore, the manuscript needs improvement in the writing of the introduction and results sections to enhance readability, although the Materials and Methods section is well-written.

Major point:

1. The authors should rewrite the introduction section. They need to include information about ZIKV, skin cells, VOCs, and the relationship between VOCs and mosquito attraction. At the end of the introduction section, the authors should clearly state the aim of the study and briefly describe how they intend to prove their hypothesis. Additionally, they should add appropriate references to support these topics, including relevant papers such as doi.org/10.1016/j.cell.2022.05.016.
2. In the results section, the authors need to include subtitles. They should also explain the rationale and purpose of each experiment in the first line of each paragraph or subtitle. Brief descriptions of the experimental methods are also necessary. For example, in the first paragraph (lines 71-82), they should provide the temporal definitions of invasion stage and the transmission stage, accompanied by appropriate references or brief descriptions of experiments, which are crucial for advancing their narrative. Additionally, the authors should verify the figure and panel numbers in this manuscript, ensuring

that the exact numbers are correctly referenced.

3. The authors performed most experiments in this study using immortalized human dermal fibroblast cells. They should not generalize their observations based on only this cell line. To confirm their observations, it is necessary to use primary human fibroblasts or several different cell lines.

4. In Figures 4 and 5, the authors analyzed gene expression and proteomes in ZIKV-infected fibroblasts. While these data present intriguing findings, they lack a clear connection between the VOCs and the identified genes or proteins. Moreover, the proteomic analysis was performed using data from ZIKV-infected human cells sourced from PubMed, rather than fibroblasts. This means that transcriptomic and proteomic analyses were not conducted on the same cells, highlighting the need for confirmation through direct experimental testing. Furthermore, a functional study, such as a loss-of-function or gain-of-function experiment, is necessary to support the conclusions presented in lines 179-180. The authors should clarify the significance of these gene expressions in lines 153-155.

Minor point:

1. The exact figure and panel numbers should be specified.
2. The manuscript should provide information on the *Anopheles gambiae sensu lato* mosquitoes used as controls.

Version 1:

Reviewer comments:

Reviewer #1

(Remarks to the Author)

The authors have addressed all of my comments to my satisfaction. I have no further comments to add.

Reviewer #2

(Remarks to the Author)

I am satisfied with the reply of the authors. In principle, all the necessary changes and experiments have been made or performed, respectively.

Point by point response to reviewers:

We sincerely thank all the reviewers for their valuable comments and suggestions. We've carefully revised the manuscript to address their feedback and made significant efforts to respond to the queries raised. We are pleased that the reviewer found our work scientifically sound and that the conclusions are largely justified. We are happy to address the suggested improvements (all additional and revised texts are highlighted in the main manuscript and supplementary information).

Reviewer #1 (Remarks to the Author):

This paper entitled "Zika virus modulates human skin cells to enhance transmission success" describes measurements of the VOCs in ZIKV-infected dermal fibroblasts in tissue culture. The authors found that upon ZIKV infection, at two timepoints (10h and 24h post-infection), various VOCs were differentially produced relative to uninfected cells. When using synthetic versions of these compounds, it was found that they altered mosquito behaviour (attraction and subsequent feeding). The authors propose that the virus is doing this to improve transmission success. The scientific methods of the paper appear sound. The conclusions are mostly justified however I believe some additional commentary about the limitations of the study are necessary, as listed below.

1. The authors state in their titles that the virus mediates human skin cells, however they just looked in in vitro models, not in actual human skin. There was also no direct evidence of enhanced transmission success in these experiments, just surrogates of transmission. Therefore I think this title could be toned down a bit to reflect this.

Authors: As suggested, we have updated the title to "Zika virus modulates **human fibroblasts** to enhance transmission success **in a controlled lab-setting**" to clearly specify the in vitro nature of the study.

2. In the same vein, there has been work demonstrating the impact of the microbiota of the skin being involved with altering mosquito attraction (e.g. <https://doi.org/10.1016/j.cell.2022.05.016>). Perhaps some discussion on this, placing the results here in the context of the microbiota on the skin, would be beneficial to the readers.

Authors: We have incorporated and discussed the highlighted reference in the introduction and discussion as requested.

Reviewer #2 (Remarks to the Author):

This manuscript suggests that ZIKV infection is modulated by VOCs in infected dermal fibroblasts and alters the emission of VOCs at the invasion and transmission stage. The authors used a human dermal fibroblast cell line to examine the VOC profile during ZIKV infection and analyzed related gene expression using RNA sequencing and proteomics. While the overall topic is interesting, weaknesses in experimental design and interpretation limit the impact of the results. Furthermore, the manuscript needs improvement in the writing of the introduction and results sections to enhance readability, although the Materials and Methods section is well-written.

Major point:

1. The authors should rewrite the introduction section. They need to include information about ZIKV, skin

cells, VOCs, and the relationship between VOCs and mosquito attraction. At the end of the introduction section, the authors should clearly state the aim of the study and briefly describe how they intend to prove their hypothesis. Additionally, they should add appropriate references to support these topics, including relevant papers such as doi.org/10.1016/j.cell.2022.05.016.

Authors: As recommended by the reviewer, we have expanded our introduction by adding information on ZIKV, skin cells, VOCs, and the relationship between VOCs and mosquito attraction. We have also clearly stated our aim and the intention behind testing our hypotheses. Additionally, we have included multiple references to support this information, including the suggested reference.

Acevedo CA, Sánchez EY, Reyes JG, Young ME. Volatile organic compounds produced by human skin cells. *Biol Res.* 2007;40(3):347-55. Epub 2008 Apr 17. PMID: 18449462.

BERNIER U, KLINE D, BARNARD D, SCHRECK E, YOST R (2000) Analysis of Human Skin Emanations by Gas Chromatography/Mass Spectrometry. 2. Identification of Volatile Compounds That Are Candidate Attractants for the Yellow Fever Mosquito (*Aedes aegypti*). *Anal Chem* 72: 747-756

Kisiel, M.A., Klar, A.S. (2019). Isolation and Culture of Human Dermal Fibroblasts. In: Böttcher-Haberzeth, S., Biedermann, T. (eds) *Skin Tissue Engineering. Methods in Molecular Biology*, vol 1993. Humana, New York, NY. https://doi.org/10.1007/978-1-4939-9473-1_6

Alberts B, Johnson A, Lewis J et al (2002) Fibroblasts and their transformations: the connective-tissue cell family. In: *Molecular biology of the cell*, 4th edn. Garland Science, New York

Musso, D. & Gubler, D. J. Zika virus. *Clin. Microbiol. Rev.* 29, 487–524 (2016).

Zhang H, Zhu Y, Liu Z, Peng Y, Peng W, Tong L, Wang J, Liu Q, Wang P, Cheng G. A volatile from the skin microbiota of flavivirus-infected hosts promotes mosquito attractiveness. *Cell.* 2022 Jun 28:S0092-8674(22)00641-9. doi: 10.1016/j.cell.2022.05.016.

2. In the results section, the authors need to include subtitles. They should also explain the rationale and purpose of each experiment in the first line of each paragraph or subtitle. Brief descriptions of the experimental methods are also necessary. For example, in the first paragraph (lines 71-82), they should provide the temporal definitions of invasion stage and the transmission stage, accompanied by appropriate references or brief descriptions of experiments, which are crucial for advancing their narrative. Additionally, the authors should verify the figure and panel numbers in this manuscript, ensuring that the exact numbers are correctly referenced.

Authors: We have added subtitles to the results section as recommended. Additionally, we included brief explanations of the rationale and experimental methods in each results subsection, as suggested. All figures and panel numbers in the manuscript have been thoroughly checked and corrected.

3. The authors performed most experiments in this study using immortalized human dermal fibroblast cells. They should not generalize their observations based on only this cell line. To confirm their observations, it is necessary to use primary human fibroblasts or several different cell lines.

Authors: To validate our findings observed in immortalised human dermal fibroblast cells, we conducted an additional experiment comparing the overall proteomes of these immortalised cells with those of two

primary fibroblast cell lines, using liquid chromatography–mass spectrometry. The results showed highly similar proteomic profiles, reducing the likelihood of artificial effects specific to immortalised fibroblasts. We have incorporated this experiment into the results, methods, and ethics sections of the manuscript (all additional information is highlighted in the main manuscript and supplementary information). The results are presented in the new Supplementary Fig. 4 (c-d), and all subsequent references have been renamed accordingly.

4. In Figures 4 and 5, the authors analyzed gene expression and proteomes in ZIKV-infected fibroblasts. While these data present intriguing findings, they lack a clear connection between the VOCs and the identified genes or proteins. Moreover, the proteomic analysis was performed using data from ZIKV-infected human cells sourced from PubMed, rather than fibroblasts. This means that transcriptomic and proteomic analyses were not conducted on the same cells, highlighting the need for confirmation through direct experimental testing. Furthermore, a functional study, such as a loss-of-function or gain-of-function experiment, is necessary to support the conclusions presented in lines 179-180. The authors should clarify the significance of these gene expressions in lines 153-155.

Authors: To gain a better understanding of the potential key players involved in VOC synthesis, we conducted a new experiment comparing the overall proteomes of ZIKV- and mock-infected immortalised human dermal fibroblasts using liquid chromatography–mass spectrometry, under the same conditions as our previous RNA-seq analysis. The RNA-seq and proteomics results are not entirely congruent; however, this does not discount the relevance of any of the identified genes, proteins, or pathways in the context of ZIKV infection. This incongruence could stem from several factors: (I) proteome analysis is inherently complex and does not capture all proteins expressed in a cell, (II) changes in RNA expression do not always translate to changes in protein abundance (e.g., compensatory changes at the protein level), and (III) protein expression is regulated through various mechanisms, including post-translational modifications (PTMs).

We have integrated this experiment into the results, methods, discussion, and ethics sections of the manuscript (all additional information is highlighted in the main manuscript and supplementary information), as well as in Extended data file 4V. The results are presented in the new Supplementary Fig. 4 (a-b), and all subsequent references have been renamed accordingly. While our proteomic analysis did not reveal any direct links between proteins and VOC synthesis, the meta-analysis remains valuable by identifying several enzymes that could be involved in VOC synthesis. Given the number of identified candidates, establishing a clear connection between VOCs and the identified genes or proteins will require further investigation in an additional study. We have also added information on the significance of altered gene expression, based on the significance thresholds defined in the methods section.

Minor point:

1. The exact figure and panel numbers should be specified.

Authors: All figure and panel numbers have been checked and corrected.

*2. The manuscript should provide information on the *Anopheles gambiae sensu lato* mosquitoes used as controls.*

Authors: Additional information on the *Anopheles gambiae sensu lato* used in the study has been added to the "Mosquito Rearing" section of the methods and result section.

Point by point response to reviewers:

We sincerely thank all the reviewers for their valuable comments and suggestions. We've carefully revised the manuscript to address their feedback and made significant efforts to respond to the queries raised. We are pleased that the reviewer found our work scientifically sound and that the conclusions are largely justified. We are happy to address the suggested improvements (all additional and revised texts are highlighted in the main manuscript and supplementary information).

Reviewer #1 (Remarks to the Author):

This paper entitled "Zika virus modulates human skin cells to enhance transmission success" describes measurements of the VOCs in ZIKV-infected dermal fibroblasts in tissue culture. The authors found that upon ZIKV infection, at two timepoints (10h and 24h post-infection), various VOCs were differentially produced relative to uninfected cells. When using synthetic versions of these compounds, it was found that they altered mosquito behaviour (attraction and subsequent feeding). The authors propose that the virus is doing this to improve transmission success. The scientific methods of the paper appear sound. The conclusions are mostly justified however I believe some additional commentary about the limitations of the study are necessary, as listed below.

1. The authors state in their titles that the virus mediates human skin cells, however they just looked in in vitro models, not in actual human skin. There was also no direct evidence of enhanced transmission success in these experiments, just surrogates of transmission. Therefore I think this title could be toned down a bit to reflect this.

Authors: As suggested, we have updated the title to "Zika virus modulates **human fibroblasts** to enhance transmission success **in a controlled lab-setting**" to clearly specify the in vitro nature of the study.

2. In the same vein, there has been work demonstrating the impact of the microbiota of the skin being involved with altering mosquito attraction (e.g. <https://doi.org/10.1016/j.cell.2022.05.016>). Perhaps some discussion on this, placing the results here in the context of the microbiota on the skin, would be beneficial to the readers.

Authors: We have incorporated and discussed the highlighted reference in the introduction and discussion as requested.

Reviewer #2 (Remarks to the Author):

This manuscript suggests that ZIKV infection is modulated by VOCs in infected dermal fibroblasts and alters the emission of VOCs at the invasion and transmission stage. The authors used a human dermal fibroblast cell line to examine the VOC profile during ZIKV infection and analyzed related gene expression using RNA sequencing and proteomics. While the overall topic is interesting, weaknesses in experimental design and interpretation limit the impact of the results. Furthermore, the manuscript needs improvement in the writing of the introduction and results sections to enhance readability, although the Materials and Methods section is well-written.

Major point:

1. The authors should rewrite the introduction section. They need to include information about ZIKV, skin

cells, VOCs, and the relationship between VOCs and mosquito attraction. At the end of the introduction section, the authors should clearly state the aim of the study and briefly describe how they intend to prove their hypothesis. Additionally, they should add appropriate references to support these topics, including relevant papers such as doi.org/10.1016/j.cell.2022.05.016.

Authors: As recommended by the reviewer, we have expanded our introduction by adding information on ZIKV, skin cells, VOCs, and the relationship between VOCs and mosquito attraction. We have also clearly stated our aim and the intention behind testing our hypotheses. Additionally, we have included multiple references to support this information, including the suggested reference.

Acevedo CA, Sánchez EY, Reyes JG, Young ME. Volatile organic compounds produced by human skin cells. *Biol Res.* 2007;40(3):347-55. Epub 2008 Apr 17. PMID: 18449462.

BERNIER U, KLINE D, BARNARD D, SCHRECK E, YOST R (2000) Analysis of Human Skin Emanations by Gas Chromatography/Mass Spectrometry. 2. Identification of Volatile Compounds That Are Candidate Attractants for the Yellow Fever Mosquito (*Aedes aegypti*). *Anal Chem* 72: 747-756

Kisiel, M.A., Klar, A.S. (2019). Isolation and Culture of Human Dermal Fibroblasts. In: Böttcher-Haberzeth, S., Biedermann, T. (eds) *Skin Tissue Engineering. Methods in Molecular Biology*, vol 1993. Humana, New York, NY. https://doi.org/10.1007/978-1-4939-9473-1_6

Alberts B, Johnson A, Lewis J et al (2002) Fibroblasts and their transformations: the connective-tissue cell family. In: *Molecular biology of the cell*, 4th edn. Garland Science, New York

Musso, D. & Gubler, D. J. Zika virus. *Clin. Microbiol. Rev.* 29, 487–524 (2016).

Zhang H, Zhu Y, Liu Z, Peng Y, Peng W, Tong L, Wang J, Liu Q, Wang P, Cheng G. A volatile from the skin microbiota of flavivirus-infected hosts promotes mosquito attractiveness. *Cell.* 2022 Jun 28:S0092-8674(22)00641-9. doi: 10.1016/j.cell.2022.05.016.

2. In the results section, the authors need to include subtitles. They should also explain the rationale and purpose of each experiment in the first line of each paragraph or subtitle. Brief descriptions of the experimental methods are also necessary. For example, in the first paragraph (lines 71-82), they should provide the temporal definitions of invasion stage and the transmission stage, accompanied by appropriate references or brief descriptions of experiments, which are crucial for advancing their narrative. Additionally, the authors should verify the figure and panel numbers in this manuscript, ensuring that the exact numbers are correctly referenced.

Authors: We have added subtitles to the results section as recommended. Additionally, we included brief explanations of the rationale and experimental methods in each results subsection, as suggested. All figures and panel numbers in the manuscript have been thoroughly checked and corrected.

3. The authors performed most experiments in this study using immortalized human dermal fibroblast cells. They should not generalize their observations based on only this cell line. To confirm their observations, it is necessary to use primary human fibroblasts or several different cell lines.

Authors: To validate our findings observed in immortalised human dermal fibroblast cells, we conducted an additional experiment comparing the overall proteomes of these immortalised cells with those of two

primary fibroblast cell lines, using liquid chromatography–mass spectrometry. The results showed highly similar proteomic profiles, reducing the likelihood of artificial effects specific to immortalised fibroblasts. We have incorporated this experiment into the results, methods, and ethics sections of the manuscript (all additional information is highlighted in the main manuscript and supplementary information). The results are presented in the new Supplementary Fig. 4 (c-d), and all subsequent references have been renamed accordingly.

4. In Figures 4 and 5, the authors analyzed gene expression and proteomes in ZIKV-infected fibroblasts. While these data present intriguing findings, they lack a clear connection between the VOCs and the identified genes or proteins. Moreover, the proteomic analysis was performed using data from ZIKV-infected human cells sourced from PubMed, rather than fibroblasts. This means that transcriptomic and proteomic analyses were not conducted on the same cells, highlighting the need for confirmation through direct experimental testing. Furthermore, a functional study, such as a loss-of-function or gain-of-function experiment, is necessary to support the conclusions presented in lines 179-180. The authors should clarify the significance of these gene expressions in lines 153-155.

Authors: To gain a better understanding of the potential key players involved in VOC synthesis, we conducted a new experiment comparing the overall proteomes of ZIKV- and mock-infected immortalised human dermal fibroblasts using liquid chromatography–mass spectrometry, under the same conditions as our previous RNA-seq analysis. The RNA-seq and proteomics results are not entirely congruent; however, this does not discount the relevance of any of the identified genes, proteins, or pathways in the context of ZIKV infection. This incongruence could stem from several factors: (I) proteome analysis is inherently complex and does not capture all proteins expressed in a cell, (II) changes in RNA expression do not always translate to changes in protein abundance (e.g., compensatory changes at the protein level), and (III) protein expression is regulated through various mechanisms, including post-translational modifications (PTMs).

We have integrated this experiment into the results, methods, discussion, and ethics sections of the manuscript (all additional information is highlighted in the main manuscript and supplementary information), as well as in Extended data file 4V. The results are presented in the new Supplementary Fig. 4 (a-b), and all subsequent references have been renamed accordingly. While our proteomic analysis did not reveal any direct links between proteins and VOC synthesis, the meta-analysis remains valuable by identifying several enzymes that could be involved in VOC synthesis. Given the number of identified candidates, establishing a clear connection between VOCs and the identified genes or proteins will require further investigation in an additional study. We have also added information on the significance of altered gene expression, based on the significance thresholds defined in the methods section.

Minor point:

1. The exact figure and panel numbers should be specified.

Authors: All figure and panel numbers have been checked and corrected.

*2. The manuscript should provide information on the *Anopheles gambiae sensu lato* mosquitoes used as controls.*

Authors: Additional information on the *Anopheles gambiae sensu lato* used in the study has been added to the "Mosquito Rearing" section of the methods and result section.